# The Association of Smoking with Contact Dermatitis: A Cross-Sectional Study

**DOI:** 10.3390/healthcare11030427

**Published:** 2023-02-02

**Authors:** Ghadah F. Alotaibi, Hanan H. Alsalman, Rama A. Alhallaf, Rayan A. Ahmad, Hashem A. Alshareef, Jumanah Meshari Muammar, Fahad M. Alsaif, Felwah F. Alotaibi, Mohamed F. Balaha, Nehad J. Ahmed, El-Sayed Khafagy, Hadil F. Alotaibi, Rahaf Alshammari, Sarah Fatani

**Affiliations:** 1Dermatology Department, College of Medicine, King Saud University, Riyadh 145111, Saudi Arabia; 2College of Medicine, Imam Muhammad ibn Saud Islamic University, Riyadh 11432, Saudi Arabia; 3College of Medicine, Almaarefa University, Riyadh 11597, Saudi Arabia; 4Department of Family Medicine, King Fahad Medical City, Riyadh 11525, Saudi Arabia; 5Clinical Pharmacy Department, College of Pharmacy, Prince Sattam Bin Abdulaziz University, Al-Kharj 11942, Saudi Arabia; 6Pharmacology Department, Faculty of Medicine, Tanta University, El-Gish Street, Tanta 31527, Egypt; 7Department of Pharmaceutics, College of Pharmacy, Prince Sattam Bin Abdulaziz University, Al-Kharj 11942, Saudi Arabia; 8Department of Pharmaceutics and Industrial Pharmacy, Faculty of Pharmacy, Suez Canal University, Ismailia 41522, Egypt; 9Department of Pharmaceutical Sciences, College of Pharmacy, Princess Nourah bint Abdulrahman University, P.O. Box 84428, Riyadh 11671, Saudi Arabia; 10Department of Pharmacy, King Fahad Specialist Hospital, Dammam 15215, Saudi Arabia

**Keywords:** allergic contact dermatitis, contact dermatitis, irritant contact dermatitis, smoking

## Abstract

Contact dermatitis is a chronic inflammatory skin disorder with a highly variable prevalence worldwide. Smoking plays a crucial role in mediating inflammatory skin conditions such as contact dermatitis. The present study aimed to investigate the association between smoking status and contact dermatitis in the Saudi population. The patients in the present study were individuals older than 18 years who were diagnosed with contact dermatitis and received a patch test at the Department of Dermatology of King Saud University Medical City from March 2003 through February 2019. All patients were interviewed by phone to complete a specific pre-designed questionnaire to assess tobacco use or exposure history. The total number of enrolled patients in the study was 308 (91 males and 217 females), all with contact dermatitis. Data from the present study suggest that the prevalence of allergic contact dermatitis in smokers may be less than that in non-smokers. Moreover, the prevalence of irritant contact dermatitis in smokers is more significant than in non-smokers. Finally, left-hand contact dermatitis is significantly associated with smoking. Therefore, there is a strong association between smoking and irritant contact dermatitis, especially in the Saudi population, regarding the left hand. Further epidemiologic studies are needed to further explore the role of smoking in the occurrence of contact dermatitis and to explore the possible mechanisms.

## 1. Introduction

Contact dermatitis (CD) is a chronic inflammatory skin disorder characterized by a skin reaction after exposure to an external agent [1]. CD is further subdivided into allergic contact dermatitis (ACD) and irritant contact dermatitis (ICD) [2]. ACD is a type-four hypersensitivity reaction that is manifested after exposure to an agent with a previous history of sensitization, while ICD is a non-specific reaction after exposure to an irritant [1,2,3]. Clinical manifestations of ACD and ICD are similar and include erythematous vesicular scaly plaques with well-defined margins that are reciprocal to the contact area [4]. The prevalence of CD is highly variable in different parts of the world. In the United States (U.S.), the prevalence was estimated to be between 1.5% and 5.4%, while, in Spain, the prevalence reached up to 28% [5,6]. In Saudi Arabia, contact dermatitis is considered a fairly common skin disorder [7].

Several factors can trigger ACD and ICD, such as the type of irritant and allergen, duration of exposure, region, age, gender, race, occupation, and atopy [8]. Smoking is considered popular globally [9]. In Saudi Arabia, the smoking prevalence is estimated to be around 12.2% [10]. CD could result from direct contact with the components of cigarettes or through the chemical effects of inhalants [11]. Armstrong et al. reported that smoking plays a role in mediating inflammatory skin conditions [12]. There are several ways in which tobacco smoking might contribute to skin inflammation [13], such as oxidative damage due to free radicals or nicotine-induced T-cell activation, pro-inflammatory cytokine release, or keratinocyte proliferation [13,14,15].

Multiple studies have focused on the possible pathogenic role of tobacco smoking in CD, and some studies have reported a relevant association [16,17,18]. However, other observations failed to identify a significant association [19,20]. From the above evidence, the association between CD and smoking has not yet been established, especially when considering the association between smoking status and allergic and irritant CD. Furthermore, to our knowledge, no studies have investigated the association between smoking and CD in Saudi Arabia.

Therefore, the present study aimed to determine the association between smoking, ACD, and ICD; describe the general characteristics of smokers with contact dermatitis; and determine whether there is an association between the duration, frequency, or type of smoking with contact dermatitis. Additionally, we sought to identify whether there is an association between the location of CD and smoking among adults in Saudi Arabia.

## 2. Materials and Methods

The present study was a retrospective cross-sectional study conducted among patients with CD. The study was conducted in accordance with the Declaration of Helsinki, followed the STROBE guidelines, and was approved by the Institutional Review Board of King Saud University College of Medicine IRB, Approval No. E-18-2858. Informed consent was obtained from all subjects involved in the study. Moreover, written informed consent has been obtained from the patients for publication. A patch test was performed on participants at the dermatology department of King Khaled University Hospital (KKUH), Riyadh, Saudi Arabia, from March 2003 through February 2019. The thin-layer rapid-use epicutaneous (TRUE) test unit contains panels 1.3 and 2.3 and includes 24 common allergens. The TRUE patch test is used in our dermatology department. In addition, patients were enrolled through the hospital’s dermatology database.

All patients were interviewed by phone to complete a pre-designed questionnaire for assessing tobacco use or exposure history. All patients ≥ 18 years old, diagnosed with CD, and receiving a patch test were enrolled in the study and divided into two groups, ACD and ICD. The main criteria for ACD diagnosis were a positive patch test for relevant contact allergens, a direct relationship with allergen exposure or avoidance, and the spreading of skin lesions. A history of excessive contact with potential irritants, the temporal course of symptoms, and a negative patch test were the major criteria for diagnosing ICD. Patients with other skin diseases (e.g., psoriasis, mycosis fungoides, and atopic dermatitis) were excluded from the study. 

The questionnaire was validated by expert academicians, who ensured that the questions effectively captured the topic under investigation. Based on the experts’ opinions, the tool’s content validity was estimated to be CVR = 0.63, CVI = 0.79. Since all questions had high CVR and high CVI, all questions were kept. The questionnaire’s reliability was calculated to be α = 0.95. The questionnaire was divided into four sections. The first section included demographic data, such as gender and educational level. The second section assessed the patient’s smoking status, including cigarette consumption and other related tobacco-containing products, and was based on a validated questionnaire (Global Tobacco Surveillance System) [21]. The third section focused on the dermatitis status of the patient, where it occurred, and whether it was related to smoking. The last section assessed atopic history. The research team developed the third and fourth sections based on the study objectives over multiple meetings. Two expert biostatisticians and a family physician revised the questions with the research team. Forward/backward translation was performed, and a pilot study was conducted on the first 30 (10%) patients, who were asked whether there was any confusion about any of the items and whether they had any suggestions for improving the items.

The data were analyzed using the Statistical Package for Social Studies (SPSS 22; IBM Corp., New York, NY, USA). The Chi-square test was used to analyze categorical variables, and Student’s t-test was used to assess continuous variables. Logistic regression was used to evaluate the risk factors. Continuous variables were expressed as mean ± standard deviation, and categorical variables were expressed as percentages. A *p*-value < 0.05 was considered statistically significant.

## 3. Results

Contact dermatitis is an inflammatory disease of the skin that is common in the general population. The pathophysiology of allergic contact dermatitis starts with the contact of the allergen with the skin [22]. The pathophysiology of contact dermatitis is shown in Figure 1.

In the present study, a total of 308 (91 males and 217 females) out of 350 patients agreed to participate in the current study, giving a response rate of 88%. Among them, 150 patients were diagnosed with ACD, and 158 were diagnosed with ICD. The total number of smokers was 31, and a significantly higher percentage of males were smokers than females (83.9% vs. 16.1%, respectively). There was no statistically significant difference between smokers and non-smokers regarding age, educational level, dry skin, allergic rhinitis, and asthma. However, the previous parameters were higher in the non-smoker group (Table 1).

Moreover, the prevalence of ACD among smokers was 50% less than among non-smokers on average (prevalence ratio (PR) = 0.5). On the other hand, smokers had a 50% higher prevalence (PR = 1.5) of ICD than non-smokers (Table 2).

After adjusting the data for age and gender using multivariable regression analysis, the effect of smoking on the prevalence of allergic contact allergy was not statistically significant (Table 3).

Furthermore, a higher percentage of patients with ICD were found to use cigarettes, cigars, and shisha compared with those with ACD, at 66.67%, 100.00%, and 66.67% vs. 33.33%, 0.00%, and 33.33%, respectively; however, the difference was only significant in the case of cigars (Table 4).

In addition, the mean smoking duration of the current study’s smokers was 12.83 ± 9.55 years, and it was longer for ACD than ICD patients, at 20.25 ± 16.92 and 11.26 ± 6.99 years, respectively, but the difference did not reach statistical significance (*p* = 0.369). These data are shown in Table 5.

Additionally, the current study’s results revealed that many smokers and non-smokers reported that the disease (either ICD or ACD) affected more than one body area. For the single affected site, the hand was the highest in the two cases (ICD and ACD) and was not significantly higher in the smoker group, at 54 (19.5) vs. 9 (29), respectively. Moreover, there were no significant differences between smokers and non-smokers regarding the affected part of the body (Table 6).

Moreover, the analysis of the data of the present study showed that the left hand was affected in a more significant proportion (18.7%) in the smoker group compared to the non-smoker group (4.9%, *p*-value = 0.03) (Table 7).

## 4. Discussion

The association between CD and smoking has not yet been established, especially in Saudi Arabia. However, Linneberg et al. reported a significant and positive dose-dependent relationship between smoking and CD, while a cross-sectional study of 520 participants found no association between smoking and nickel and cobalt sensitization [16,17]. 

The current study investigated the possible association between tobacco smoking and ACD and ICD. We found that ICD was more frequent among smokers. These results could be explained by the alteration in barrier function in tobacco smoking, allowing irritants and allergens to cross the barrier easily [18]. Moreover, our results have shown that the risk of ACD, which is mediated by Th1, was actually low in tobacco smokers. Tobacco smoking favors a Th1-mediated immune response and suppresses the Th2-mediated immune response [17]. However, it could be argued that the human immune system is very complex, and Th1/Th2-mediated immune responses partially explain the development of various immune responses [19]. The previous pathogenesis might support our result regarding the lack of a significant association between allergic rhinitis and asthma (Th-2-mediated diseases) with smoking (favoring Th1-mediated responses) [13].

Moreover, two prospective population-based studies have suggested that tobacco smoking might decrease the risk of IgE-mediated diseases [17]. In addition, tobacco products have many irritants and sensitizing agents that vary depending on the manufacturer [20]. This might explain why only one-hand (left-hand) dermatitis tended to be more frequent among smokers [23]. However, our questionnaire did not specify the location of the contact dermatitis lesions, whether it was in the smoking hand or the other. Furthermore, our results showed no significant differences between smoking status and skin dryness, nor did we detect a relationship with the location of contact dermatitis.

In a review article addressing the association of smoking with contact dermatitis, seven out of eight articles described a positive relationship between smoking and ACD or ICD, and the authors concluded that smoking might be an essential risk factor for both ACD and ICD [17]. Another systematic review and meta-analysis found that allergic dermatitis was associated with active and passive smoking [24]. These results agreed with our results regarding ICD and were in contrast to the current study’s results regarding ACD.

Sawada et al. reported that smoking increases the risk of hand eczema and contact dermatitis. Although the exact mechanism is still somewhat debatable, smoking boosts the Th1 immune response and IFN- cytokine production [25]. Patients with mixed allergic and irritant hand eczema, hyperhidrosis, and pertinent contact sensitizations are more likely to be smokers, according to Molin et al. [26]. Moreover, Kantor et al. reported that both active and passive smoking are linked to an increased prevalence of atopic dermatitis [27]. According to Jing et al., second-hand smoke exposure is an independent but modifiable risk factor for atopic dermatitis and hand eczema in teenagers [28]. Strzelak et al. stated that the Th2/Th17 polarization highlights how smoking can encourage the onset of atopic dermatitis. Furthermore, contact dermatitis and cigarette smoke have a substantial correlation [15].

It has been reported that tobacco and tobacco smoke are strongly associated with different skin conditions; among them, CD is of prime importance [11]. CD often occurs in tobacco harvesters, curers, and cigar makers, while, in contrast, it rarely affects smokers and only exceptionally cigarette packaging workers [20]. The sensitizing compound in tobacco remains unknown: nicotine, while highly toxic, does not seem to cause sensitization, except in rare cases [29]. In addition to natural substances, several compounds are added to tobacco during its processing and manufacture, so identifying the etiological factors is difficult [30].

The main limitation of the study was that the data were subjective, because they were collected through phone calls. The second limitation was that, due to the small sample size and because the study was only conducted in the Saudi Arabian population, it is not possible to draw generalized conclusions for other countries. Further studies are needed to confirm the relationship between smoking and dermatitis.

## 5. Conclusions

The present study revealed that there is a strong association between smoking and irritant contact dermatitis. On the other hand, the study showed that smoking is inversely associated with allergic contact dermatitis. Additional epidemiologic studies are needed to further explore the role of smoking in the occurrence of contact dermatitis and to explore the possible mechanisms.

## Figures and Tables

**Figure 1 healthcare-11-00427-f001:**
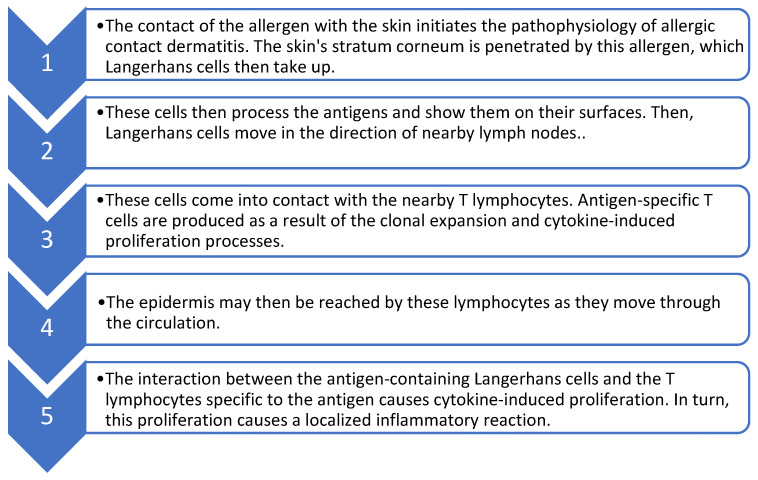
The pathophysiology of contact dermatitis [22].

**Table 1 healthcare-11-00427-t001:** Demographic data of the patients and their associated disorders.

	Smoker N (%)	Non-Smoker N (%)	*p*-Value
Male	26 (83.9)	65 (23.5)	<0.001
Female	5 (16.1)	212 (76.5)	
Age (Mean ± SD)	38 ± 5.6	40 ± 6.7	>0.05
Educational Level	19 (61.3)	159 (57.4)	>0.05
Dry Skin	20 (64.5)	221 (79.8)	>0.05
Allergic Rhinitis	11(35.5)	137(49)	>0.05
Asthma	9 (29)	129 (46)	>0.05

**Table 2 healthcare-11-00427-t002:** Effect of smoking on the prevalence of ACD and ICD.

	Smokers N (%)	Non-Smokers N (%)	Relative Risk	*p*-Value
ACD	8 (25.8)	142 (51.3)	0.5	0.007
ICD	23 (74.2)	135 (48.7)	1.5	

**Table 3 healthcare-11-00427-t003:** Effect of smoking on the prevalence of ACD after adjustment for age and gender.

	Prevalence of ACD (%)	Odds Ratio * (95% CI)	*p*-Value
Smoking Status:			
Never	51.3%	1.00	
Daily	30.0%	1.39 (0.44–4.46)	0.577
Less than daily	18.2%	0.60 (0.11–3.33)	0.563
Previously	20.6%	1.01 (0.35–2.98)	0.973

* Adjusted for age and gender.

**Table 4 healthcare-11-00427-t004:** Relation of tobacco smoking type with ICD and ACD.

	ACD		ICD		Total	*p*-Value
	N	%	N	%	N	
Cigarettes	4	33.33	8	66.67	12	0.077
Cigars	0	0.00	7	100.00	7	0.003
Shisha	3	33.33	6	66.67	9	0.127

**Table 5 healthcare-11-00427-t005:** The association between mean smoking duration and either ACD or ICD.

			ACD		ICD		*p*-Value
	Mean	SD	Mean	SD	Mean	SD	
Duration of Smoking	12.83	9.55	20.25	16.92	11.26	6.99	0.369

**Table 6 healthcare-11-00427-t006:** Relation between smoking and the affected part of the body by CD.

	Smokers N (%)	Non-Smokers N (%)	*p*-Value
Hand	9 (29)	54 (19.5)	>0.05
Face	1 (3.2)	17 (6.1)	>0.05
Foot	0	15 (5.4)	>0.05
Trunk or Other	6 (19.4)	35 (16.2)	>0.05
More Than One of the Above	15 (48.4)	132 (47.7)	>0.05

**Table 7 healthcare-11-00427-t007:** Effect of smoking on hand associated with CD.

	Smokers N (%)	Non-Smokers N (%)	*p*-Value
Right hand	3 (18.7)	14 (17.3)	>0.05
Left hand	3 (18.7)	4 (4.9)	0.03
Both	10 (62.5)	63 (77)	>0.05

## Data Availability

Not applicable.

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
