# Peer review of "The Association of Smoking with Contact Dermatitis: A Cross-Sectional Study"

_healthcare, 2023, doi:10.3390/healthcare11030427_

Round 1

Reviewer 1 Report

The current manuscript regards a cross sectional study assessing the association between smoking and contact dermatitis. It is overall interesting, produces predictable results, and the methodology is reasonably sound. Before publication, I would like to see the following alterations:

- Regarding the pathophysiology of contact dermatitis and contact dermatitis, provide a figure regarding the molecular aspects of it (what happens on a tissue/cellular level);

- Comment on the limitations of the current study, in what concerns the number of participants, the way information was collected (through phone call), and the fact that the study was only conducted in Saudi Arabia population, hence not being possible to draw generalized conclusions for other countries.

Author Response

we add a figure about the pathophysiology of contact dermatitis and contact dermatitis

we add study limitations

Reviewer 2 Report

An interesting topic, but for the publication of the article, a series of improvements are needed:

- the validation of the questionnaire used in the study must be presented;

- the discussions are insufficient, they require improvements and additional correlations

- the conclusions must be modified, the importance of the study carried out within the topic addressed, the perspective of using the results obtained should be presented

Author Response

we modified the discussion section

we modified the conclusion section

we write a paragraph about the validation of the questionnaire

Reviewer 3 Report

Congratulations for your work so far. The topic of the article is interesting and it fits within the scope of MDPI Healthcare.

The experimental and analysis methodology are suitable with high feasibility.

You have highlighted the aims, significance and the novelty of your work.

It is good that you mentioned important articles in the bibliography (Jacobsen G., Zimmer K.A., Lukács J. etc.), but it is necessary to refer your study to other reference studies in the medical literature:

1.      Sawada, Y.; Saito-Sasaki, N.; Mashima, E.; Nakamura, M. Daily Lifestyle and Inflammatory Skin Diseases. Int. J. Mol. Sci. 2021, 22, 5204. https://doi.org/10.3390/ijms22105204.

2.      Strzelak, A.; Ratajczak, A.; Adamiec, A.; Feleszko, W. Tobacco Smoke Induces and Alters Immune Responses in the Lung Triggering Inflammation, Allergy, Asthma and Other Lung Diseases: A Mechanistic Review. Int. J. Environ. Res. Public Health 2018, 15, 1033. https://doi.org/10.3390/ijerph15051033.

3.      Galiniak, S.; Rachel, M. Fractional Exhaled Nitric Oxide in Teenagers and Adults with Atopic Dermatitis. Adv. Respir. Med. 2022, 90, 237-245. https://doi.org/10.3390/arm90040033.

4.      Kantor R, Kim A, Thyssen JP, Silverberg JI. Association of atopic dermatitis with smoking: A systematic review and meta-analysis. J Am Acad Dermatol. 2016 Dec;75(6):1119-1125.e1. doi: 10.1016/j.jaad.2016.07.017. Epub 2016 Aug 16. PMID: 27542586; PMCID: PMC5216172.

5.      Molin, S., Ruzicka, T. and Herzinger, T. (2015), Smoking is associated with combined allergic and irritant hand eczema, contact allergies and hyperhidrosis. J Eur Acad Dermatol Venereol, 29: 2483-2486. https://doi.org/10.1111/jdv.12846.

6.      Jing, D., Li, J., Tao, J. et al. Associations of second-hand smoke exposure with hand eczema and atopic dermatitis among college students in China. Sci Rep 10, 17400 (2020). https://doi.org/10.1038/s41598-020-74501-2.

Please improve the references if you consider it appropriate.

Author Response

I add the recommended references 

Round 2

Reviewer 2 Report

No validation of the questionnaire was done. It is not enough to put a sentence without content. I would like you to present how the questionnaire's content validity indexes were determined, namely: Content Validity Ratio (CVR) and Content Validity Index (CVI), as well as the determination of the questionnaire's reliability using the Cronbach alpha coefficient. Without this data, the questionnaire is ineffective. Study in more detail how to use a questionnaire.

Author Response

we add the validation of the questionnaire